# Relationship between Patient Preferences, Attitudes to Treatment, Adherence, and Quality of Life in New Users of Teriflunomide

**DOI:** 10.3390/ph15101248

**Published:** 2022-10-11

**Authors:** Daniela Štrosová, Jan Tužil, Barbora Velacková Turková, Barbora Filková Pilnáčková, Lada Lžičařová de Souza, Helena Doležalová, Michaela Rašková, Michal Dufek, Tomáš Doležal

**Affiliations:** 1Value Outcomes Ltd., Vaclavska 316/12, 12000 Prague, Czech Republic; 2Medical Informatics, First Medical Faculty, Charles University, Kateřinská 1660/32, 12000 Prague, Czech Republic; 3Sanofi, Evropská 846/176a, 16000 Prague, Czech Republic; 41st Department of Neurology, St. Anne’s University Hospital, Pekařská 664/53, 60200 Brno, Czech Republic; 5Pharmacology Department, Faculty of Medicine, Masaryk University, Kamenice 753/5, 62500 Brno, Czech Republic

**Keywords:** multiple sclerosis, teriflunomide, adherence, quality of life, MMAS-8, BMQ, SEAMS

## Abstract

**Background:** A poor patient adherence often limits the real-world effectiveness of an oral disease-modifying therapy (DMT) for multiple sclerosis (MS). In the present study, we aimed to map patient preferences, attitudes toward treatment, and quality of life to identify the predictors of non-adherence to teriflunomide. **Methods:** This was a single-arm, non-interventional, multicenter study (Czech Act 378/2007 Coll.) consisting of three visits: the first at treatment initiation (teriflunomide 14 mg), and then after 3 and 9 months of therapy. We enrolled both DMT-naïve and patients who had undergone a DMT diagnosed with a clinically isolated syndrome (CIS) or relapsing-remitting multiple sclerosis (RRMS). The functional status and MS activity were estimated using the Expanded Disability Status Scale (EDSS) and annualized relapse rate (ARR); the quality of life via the Multiple Sclerosis Impact Scale (MSIS-29); the medication adherence with the Morisky Medication Adherence Scale (MMAS-8); the confidence in the ability to take medications by the Self-Efficacy for Appropriate Medication Score (SEAMS); and the attitude to the therapy via the Beliefs about Medicines Questionnaire (BMQ). After nine months of therapy, we predicted the adherence to teriflunomide (MMAS-8) by fitting a multivariate ordinal logistic model with EDSS changes, gender, previous DMT, MSIS-29, BMQ, and SEAMS as the explanatory variables. **Results:** Between 2018 and 2019, 114 patients were enrolled at 10 sites in the Czech Republic. The mean age was 41.2 years, 64.8% were diagnosed with a CIS, 52.4% were DMT-naïve, and 98.1% of patients preferred an oral administration at the baseline. The mean EDSS baseline was 1.97 and remained constant during the 9 months of therapy. The ARR baseline was 0.72 and dropped to 0.19 and 0.15 after 3 and 9 months, respectively. Despite a more than 4-fold higher ARR baseline, the treatment-naïve patients achieved an ARR at 9 months comparable with those previously treated. There were ten non-serious adverse reactions. After nine months of teriflunomide therapy, 63.3%, 21.2%, and 15.4% of patients had a high, medium, and low adherence, respectively, as per the MMAS-8; 100% of patients preferred an oral administration. The SEAMS score (odds ratio (OR) = 0.91; *p* = 0.013) and previous DMT (OR = 4.28; *p* = 0.005) were the only significant predictors of non-adherence. The disability, the quality of life, and beliefs about medicines had no measurable effect on adherence. **Conclusion:** After nine months of teriflunomide therapy, both the disability and quality of life remained stable; the relapse rate significantly decreased, 63.3% of patients had a high adherence, and 100% of patients preferred an oral administration. A low adherence was associated with previous DMT experiences and a low self-efficacy for the appropriate medication (i.e., the confidence in one’s ability to take medication correctly).

## 1. Introduction

Multiple sclerosis (MS) is a chronic progressive autoimmune disease of the central nervous system. The global prevalence of MS is estimated at 30 per 100,000; it is more frequent in females and those living at higher latitudes [1,2]. The pathological mechanism of MS involves inflammation, demyelination, and axonal and neuronal losses [3,4]. Typical symptoms include fatigue, impaired motor skills, blurred vision, bladder and bowel dysfunctions, and cognitive impairments [5]. MS is characterized by recurrent relapses. The onset typically occurs in adults at the beginning of their productive age and ultimately progresses to a severe disability [3]. MS significantly decreases the quality of life, with reported utilities ranging between 0.31 and 0.78. Patients are typically burdened with pain, discomfort, and a severe impairment in everyday activities [6].

Teriflunomide, the active metabolite of leflunomide, is an immunomodulatory agent that reduces the proliferation of rapidly dividing cells by the reversible inhibition of mitochondrial dihydroorotate dehydrogenase [7] whilst preserving the replication of slowly dividing cells that use the exogenous supplies of pyrimidine nucleotides through salvage pathways [8]. The therapeutic effect of teriflunomide in MS is mediated by a reduced number of lymphocytes, although the exact mechanism is not fully understood [7]. The efficacy and safety of teriflunomide in adults were established in three pivotal phase III trials that showed teriflunomide at doses of 7 and 14 mg was superior to a placebo in reducing relapse rates, limiting the disability progression, and reducing the MRI evidence of disease activity in patients with relapsing-remitting multiple sclerosis (RRMS) [9,10]. It also reduced the risk of relapses or new MRI lesions in patients with a clinically isolated syndrome (CIS), which is suggestive of multiple sclerosis [11].

Non-persistence and a poor adherence are common issues among MS patients taking DMTs. Non-adherence to or non-persistence with DMTs are associated with worse clinical outcomes such as frequent relapses and the disease progression [4]. On the other hand, a better adherence is associated with a better quality of life, fewer hospitalizations and emergency room visits, decreased neuropsychological issues, fewer workdays lost to MS, and lower medical costs [4]. Nonetheless, the average patient only uses the medication 76.5% of the time during the first year and 25.4% of patients completely discontinue the therapy [4].

Patient preferences and attitudes play a significant role in treatment decision-making and efforts to avoid future non-adherence [4,12,13]. Importantly, all published studies have found that oral DMTs are preferred over injectables [13]; better compliance and persistence have been reported in patients who were initially indicated for oral DMTs compared with injectables [14,15]. Teriflunomide is a once-daily oral DMT prescribed either as a first-line or as an alternative for patients with “needle fatigue” [16]. Of the oral DMTs, teriflunomide is likely to have a better persistence and adherence than dimethyl fumarate [14,15].

The primary objective of the present study was to describe the relationship between the preferences, attitudes toward treatment, adherence, and quality of life in patients initiating a teriflunomide treatment, both treatment-naïve and previously treated, in an actual clinical practice.

## 2. Results

Between March 2018 and April 2019, we enrolled 114 patients at 10 sites in the Czech Republic. A total of 105 patients (92.1%) completed the 9-month follow-up (Figure 1). The cohort was balanced in terms of gender (56.2% females); the mean age at the initiation of the teriflunomide treatment was 41.2 years. The mean BMI was 25.1, indicating a normal weight (Table 1).

Of the enrolled cohort, 68 patients (64.8%) were diagnosed with a CIS and 37 patients (35.2%) had RRMS. A total of 50 patients (47.6%) had received at least 1 previous DMT, which, on average, lasted for 3.8 years.

The most frequent previous DMT was interferon beta 1a (46.0%), followed by glatiramer acetate (34.0%) (Appendix A). The mean time from the diagnosis to the treatment was 3.5 years; it was 7.2 years for patients with a previous DMT and 0.2 years for DMT-naïve patients.

### 2.1. Primary Objective

The primary aim was explored using a sample of 82 patients who had all 4 scores available after 9 months of teriflunomide therapy. This model did not include the patient preferences regarding the route of administration because at the third visit all patients reported a preference for a peroral administration. In the final model, we predicted the adherence after nine months as per the MMAS-8 (high, medium, and low), SEAMS, BMQ (accepting, ambivalent, indifferent, and skeptical), MSIS-29 total score, gender (male/female), previous DMT (no, yes), and change in EDSS (between the baseline and visit 3) as the explanatory variables. An experience with a previous therapy had a significant negative impact on the adherence at nine months with an adjusted OR = 4.66 (95% CI: 1.67–13.04; *p* = 0.003). The effect size was additionally confirmed in both the second and the third visit by fitting a multilevel ordered logistic model (details not presented). There was also a drop in adherence observed with each other line of therapy preceding teriflunomide (details not presented). Each point increase in the SEAMS (i.e., the confidence in one’s ability to take medication correctly) improved the adherence (OR = 0.92; 95% CI: 0.86–0.99; *p* = 0.032). A change in disability, gender, quality of life, and beliefs about medicines had no effect on the patient adherence to teriflunomide (Table 2).

### 2.2. Disease Activity and Disability

As the lengths of the evaluated periods were different (i.e., three months vs. six months), the relapse rate was annualized to allow for comparisons. The baseline value of the ARR was 0.72 (95% CI: 0.57–0.91); after 3 and 9 months of treatment, it decreased to 0.19 (95% CI: 0.12–0.29) and 0.15 (95% CI: 0.09–0.25), respectively. The relative decrease from the baseline was 73.7% and 78.9% after 3 and 9 months, respectively (Table 3; Figure 2).

In patients diagnosed with a CIS and RRMS, the baseline ARR was 0.78 and 0.62, respectively; the relative decrease after 9 months was 81.1% for a CIS and 73.9% for RRMS (Appendix A). Despite a more than four-fold higher ARR baseline, after nine months of treatment, the naïve patients had an ARR comparable with those who had been previously treated (Appendix A).

The mean (± SD) EDSS baseline score was 1.97 ± 0.99. During the analyzed period, it remained comparable with the baseline at 1.97 ± 1.04 after 3 months and 2.03 ± 0.99 after 9 months of therapy (Table 3; Appendix A). The median value was equal to 2.0 at all three visits. In patients diagnosed with a CIS and RRMS, the baseline EDSS scores were 1.75 and 2.36, respectively, and remained constant during all 9 months of therapy (Appendix A).

A total of 62.9% of patients experienced at least 1 relapse during the 12 months preceding the teriflunomide initiation; 29.5% of the patients were hospitalized due to the relapse. After three months of therapy, the proportion of patients having at least one relapse decreased to 3.8%; none were hospitalized (Table 3). After nine months of therapy, the proportion increased slightly to 7.6%, with 1.0% hospitalized.

### 2.3. Quality of Life and Adherence

The mean disease-specific quality of life measured by the MSIS-29 was 26.6 at the baseline, 25.6 after 3 months, and 25.5 after 9 months of treatment. The trend was mainly driven by the psychological subscale, which decreased from 31.8 to 29.7 after 9 months of treatment. The increasing trend in the disease-specific quality of life was most pronounced in the previous DMT group; notably, the psychological subscale. These patients, however, had an already significantly worse quality of life at the baseline (*p* = 0.035) and thus did not reach the MSIS-29 score of the naïve patients, even after an improvement at the second (*p* = 0.006) or third visit (*p* = 0.038) (Appendix A). The physical subscale remained stable during the study period (21.4 at the baseline and 21.3 after 9 months) (Table 3; Appendix A).

The mean adherence score of the patients measured using the MMAS-8 was 7.6 after 3 months and 7.4 after 9 months of treatment. After nine months of the teriflunomide therapy, 63.5%, 21.2%, and 15.4% of patients were classified as having a high, medium, and low adherence using the MMAS-8, respectively (Table 3; Appendix A). The percentage of medication doses that the patients had taken in the previous three months was measured using a VAS. Patient-reported adherences were 96.2% at three months (visit 2) and 95.6% at nine months (visit 3) (Table 3); the VAS moderately correlated with the MMAS-8 (Spearman coefficient of 0.598; *p* < 0001).

Patients who were non-adherent after nine months of treatment with teriflunomide (MMAS-8 ≤ 6 points) had, on average, a lower disease-specific quality of life score compared with the adherent patients (MMAS-8 > 6 points), irrespective of the previous therapy. The same was observed when the adherence was categorized into high, medium, and non-adherent (Appendix A; Figure 3b); however, this difference was not statistically significant.

### 2.4. Attitudes toward Treatment and Preferences

The patient self-efficacy in the medication used (assessed using the SEAMS) was 28.5 after 9 months of treatment. As for beliefs about medicines (measured using the BMQ) 60.6% of patients could be classified as accepting, 31.3% as ambivalent, and 8.2% as indifferent or skeptical (Table 3). In total, 98.1% of patients preferred an oral administration over an injection at the baseline. After three and nine months of treatment, 100.0% of patients preferred an oral administration (Table 3; Appendix A).

### 2.5. Safety

Out of the 114 patients enrolled in this study, most patients (99; 86.8%) did not report any adverse events (AE). Thirteen patients (11.4%) experienced one AE and two patients (1.8%) experienced two AEs. None of the AEs were severe. Of the 17 reported AEs, the causality was established in 10 cases (1 with peripheral neuropathy, 6 with ALT elevation, 2 with an intolerance, and 1 with hair thinning). These 17 AEs occurred in 6 DMT-naïve and 11 previously treated patients. There were no cases of progressive multifocal leukoencephalopathy. No pregnancies were reported. A total of 4 patients (3.5%) discontinued the therapy.

## 3. Discussion

In our cohort, the disability remained stable through the nine months of the teriflunomide therapy. Although teriflunomide is generally considered to be a DMT with a “medium” efficacy level [17], the ARR decreased to 0.15 (95% CI: 0.09–0.25). This was in line with observations from cohorts in Denmark where the ARR dropped to 0.20 (95% CI: 0.17 to 0.22) [18] and Australia (ARR 0.22 (95% CI: 0.18 to 0.26)) [18]. The reduction in the relapse rate was even more pronounced in the treatment-naïve patients. The quality of life remained comparable with the baseline, which was in line with the results previously published for Swedish [19] and French [20] cohorts. The relationship between the disease-specific quality of life and patient adherence suggested that the adherent patients had a higher quality of life (Figure 3a; Appendix A).

To date, few studies have assessed the real-world adherence to teriflunomide and other DMTs [5,14,15,21,22,23] and none have addressed the reasons for the discontinuation of teriflunomide [24]. Our study is the first to gather data from new teriflunomide users with a focus on the predictors of adherence. A recent study from Finland showed that, among patients exclusively on teriflunomide, males and older patients were more likely to persist with the therapy [15]. In our study, we observed no relationship between the gender and adherence; however, we did observe an important relationship between self-efficacy in the medication used and adherence.

In our cohort, most switchers initiated teriflunomide after using an injectable DMT; i.e., interferons or glatiramer acetate (Appendix A). A previous experience with at least one DMT treatment before the initiation of teriflunomide was found to negatively impact the adherence. The magnitude of this effect was large; the odds for a worse adherence were more than four-fold higher in previously treated patients (Figure 3b). This was in line with previous publications showing that first- and second-line patients differed in their beliefs about the disease and the effectiveness of medicines [23]. Although we did not collect the reasons for treatment switching, we assumed that the reason for treatment switching for those with a previous DMT experience was that the previous experience was negative with regard to efficacy, safety, or tolerability [25]. A recent systematic review [13] showed that, most often, patients switched DMTs due to poor tolerability, adverse events, or at the request of a healthcare provider. Patients can be switched from injectables to oral treatments because of intolerances, increased disease activity, or simply because oral DMTs are newly available [13]. So-called “needle fatigue” and injection-related side effects were the main reasons for switching in numerous studies [25]. Contrary to our results, one would expect that patients switching from previous therapies would be more adherent to teriflunomide, which is a safe and efficacious DMT. A possible explanation for the non-adherence in the second-line could be the need for a daily administration of an oral drug being difficult to remember, especially in a patient used to a weekly administration of an injectable drug. On the other hand, the SEAMS score was not significantly different between the switchers and naïve patients at the baseline (*p* = 0.09), nor the third visit (*p* = 0.465).

Another possible cause of a lower adherence in the previously treated patients may have been the amount of time elapsed since the first onset of symptoms. A better adherence in the naïve patients might be linked to the fact that they were only recently diagnosed with MS [25].

The reasons for a worse adherence in the switchers remains unexplained by our study. Future studies would benefit from a second arm in which the same parameters were collected from patients receiving a different oral drug indicated as a second-line treatment of MS (i.e., dimethyl fumarate). This may help identify whether a lower adherence is due to an oral administration or other factors linked to teriflunomide.

## 4. Materials and Methods

### 4.1. Study Design and Standards

In accordance with Czech legal definitions, this was a single-arm, non-interventional, multicenter study (Act 378/2007 Coll.). The study consisted of 3 visits: the first at teriflunomide initiation (baseline); after 3 months ± 30 days (visit 2); and finally, after 9 months ± 60 days of treatment (visit 3). Patients unwilling or unable to complete the study visits in the specified window were considered lost to follow-up. The data were collected from patient medical records and patient questionnaires using anonymized electronic case report forms.

All patients signed informed consent regarding study participation and personal data processing. Local ethics committees approved the project at the 10 participating hospital centers under the following identifiers: KH/04/2018 (Faculty Hospital Kralovske Vinohrady); 6/18 (Hospital Ceske Budejovice); 3/18 (Faculty Hospital Olomouc); 12JS/2018 (Faculty Hospital U Sv. Anny Brno); I/18/1 (Nemocnice Teplice); L-18-05 (Institute for Clinical and Experimental Medicine); 201801 S06O (Faculty Hospital Hradec Kralove); TERIFLO8851 (Regional Hospital Tomase Bati Zlin); 762 (Nemocnice Jihlava); and 58/2018 (Faculty Hospital Ostrava). The abbreviated protocol was prospectively registered in the State Institute for Drug Control in the Czech Republic (SUKL) national database under the identifier 1801010000. Regular distant and on-site monitoring visits were conducted. We reported in line with the STROBE statement [26].

### 4.2. Subject Eligibility

The enrolment criteria included: (1) signed written informed consent; (2) aged ≥ 18 and ≤65 years; (3) a diagnosis of a CIS or RRMS; (4) eligible for an Aubagio^®^ treatment according to the product information (7) and SUKL criteria; and (5) either naïve or previously treated with a disease-modifying treatment (DMT). We excluded patients contraindicated to Aubagio^®^ as per the product information and patients not willing to fill out or unable to understand the patient information and questionnaires.

### 4.3. Parameters

The impact of MS was estimated using the Expanded Disability Status Scale (EDSS; 0 to 10) [27] and the presence of relapses at each visit. The disease-specific quality of life was assessed using the Multiple Sclerosis Impact Scale at each visit (MSIS-29; 0 to 100; higher values indicate a more significant impact of MS on daily functions, i.e., poorer health) [28]. For the medication adherence, the Morisky Medication Adherence Scale was used (MMAS-8; 0 to 8; higher values indicate a better adherence; categorization: high (MMAS-8 = 8), medium (MMAS-8 > 6), and low (MMAS-8 ≤ 6)) [29]. Additionally, a visual analog scale (VAS; 0 to 100 representing the percentage of medication doses patients had taken in the previous 3 months) was used; these assessments were conducted at 3 and 9 months. The confidence in one’s ability to perform a given task such as taking medication [30] was evaluated using the Self-Efficacy for Appropriate Medication Score (SEAMS; 13 to 39; higher values indicate higher patient self-efficacy in medication use) [31] at nine months. The attitude to treatment was evaluated by the Beliefs about Medicines Questionnaire (BMQ; categories: accepting, ambivalent, indifferent, and skeptical) [32] at nine months.

### 4.4. Statistical Analysis

The patients who continued the therapy for nine months and completed the three study visits within the defined timeframe were considered to be the per protocol population (PP). The mean ARR was calculated as the total relapses in a given period divided by the total patient-years at risk; we used the exact Poisson confidence interval (CI). An analysis of the primary objective was performed by fitting the univariate and multivariate ordinal logistic models; the score for adherence (MMAS-8) was predicted using scores representing the patient’s quality of life (MSIS-29), attitudes toward treatment (SEAMS and BMQ), and patient preferences for the dosage form (i.e., oral vs. injectable). All parameters were assessed after nine months of treatment (at the third visit). The analysis of the primary objective included all per protocol (PP) patients who had scores from the given questionnaires. The data were analyzed using STATA 15.0 software (StataCorp LLC, College Station, TX, USA).

## 5. Conclusions

After nine months of a teriflunomide therapy, both the mean disability and the mean quality of life remained stable; the relapse rate significantly decreased. Despite a more than four-fold higher ARR baseline, the treatment-naïve patients achieved an ARR at nine months comparable with those previously treated. After nine months, 63.3% of patients had a high adherence, and 100% of patients preferred an oral administration. A low adherence was associated with previous DMT experiences and a low self-efficacy for appropriate medication scores (i.e., the self-reported confidence in one’s ability to take medication correctly). These findings should be confirmed with large evidence-based observational studies.

## Figures and Tables

**Figure 1 pharmaceuticals-15-01248-f001:**
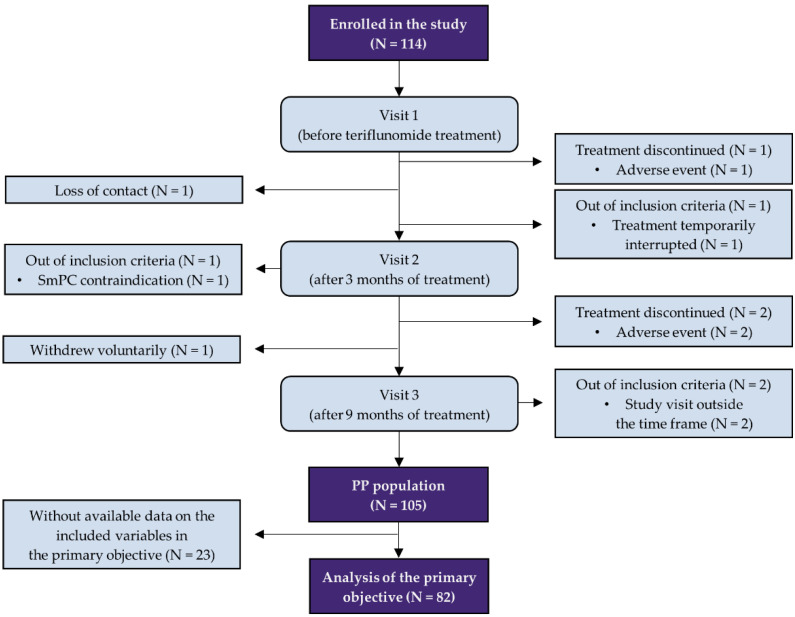
Patient flow according to the STROBE statement. Of the 114 patients enrolled, 9 (7.9%) were excluded from the analysis. The remaining 105 patients represent the per protocol (PP) population presented in the results. For the analysis of the primary objective, only patients who had available data on the included variables (i.e., all questionnaires completed) after 9 months of treatment were included (N = 82).

**Figure 2 pharmaceuticals-15-01248-f002:**
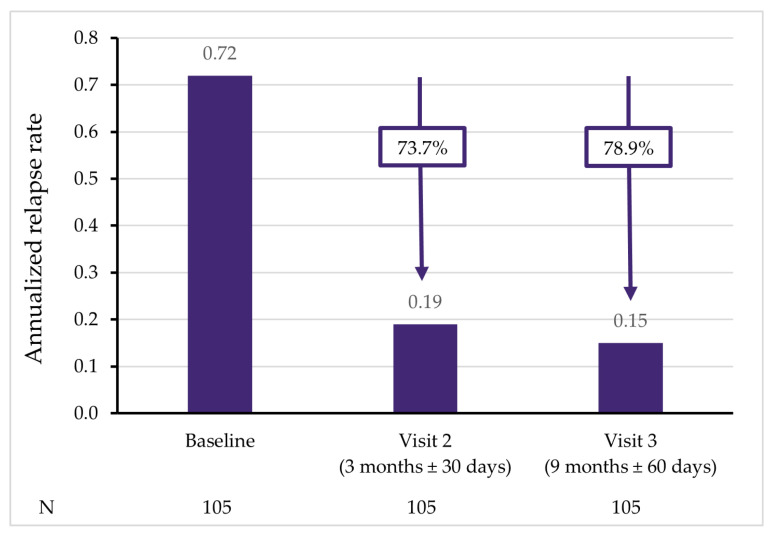
The annualized relapse rate after 3 and 9 months of teriflunomide therapy.

**Figure 3 pharmaceuticals-15-01248-f003:**
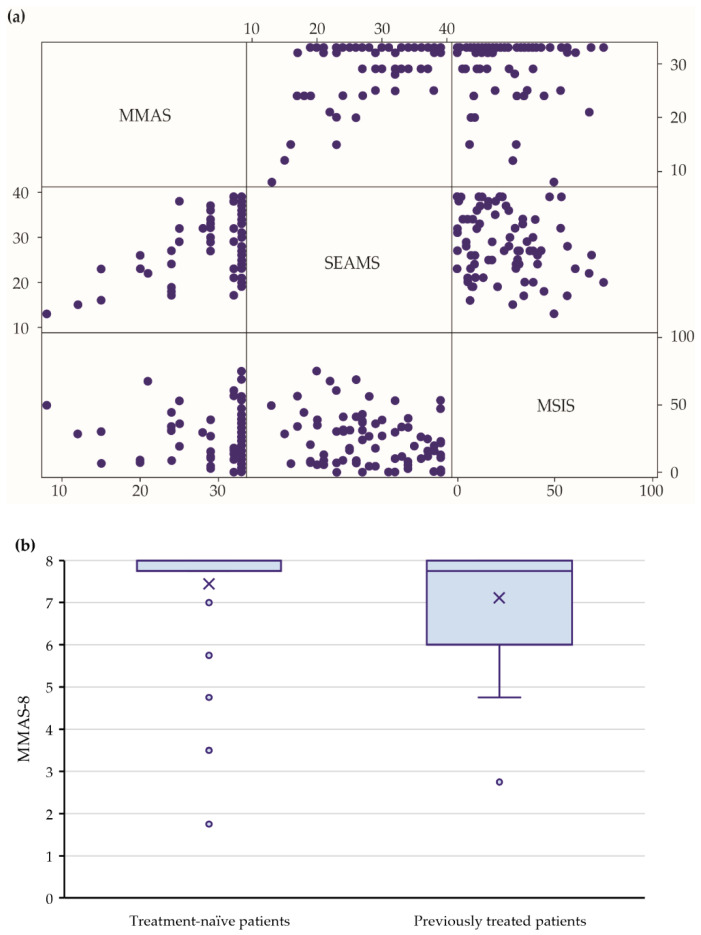
Questionnaire scores relevant to the primary aim after 9 months of treatment (N = 82). (**a**) Matrix showing observed two-dimensional relationships between patient adherence (MMAS-8), Self-Efficacy in Medication Use (SEAMS), and quality of life (MSIS-29) at 9 months (visit 3). (**b**) The difference in adherence (MMAS-8) between treatment-naïve and previously treated patients at 9 months (visit 3).

**Table 1 pharmaceuticals-15-01248-t001:** Baseline characteristics of the cohort before teriflunomide initiation.

Baseline Characteristics	Mean (±SD)/Number (%)
Number of patients	105
Female	59 (56.2%)
Age (years)	41.2 (±10.6)
Diagnosis of CIS	68 (64.8%)
Diagnosis of RRMS	37 (35.2%)
Time from diagnosis (years)	3.5 (±5.5)
Time from diagnosis for previously treated patients (years) ^1^	7.2 (±6.1)
Time from diagnosis for DMT-naïve patients (years) ^1^	0.2 (±0.2)
DMT-naïve patients	55 (52.4%)
One previous DMT	37 (35.2%)
Two or more previous DMTs	13 (12.4%)
Duration of previous treatment (years) ^2^	3.8 (±4.5)
Baseline BMI	25.1 (±4.4)

^1^ Previously treated patients (N = 50) and DMT-naïve patients (N = 55). ^2^ Only for previously treated patients.

**Table 2 pharmaceuticals-15-01248-t002:** Ordinal logistic regression models for the prediction of adherence category (MMAS-8; low, medium, and high) (N = 82). The pseudo-R^2^ of the final multivariate model was 11.13%, which suggested that the concept of adherence was influenced by other important factors that were not measured in our setting.

MMAS-8 [Decreasing] ^1^	Unadjusted ^2^		Adjusted ^2^		
	OR (95% CI) ^3^	*p*-Value ^3^	OR (95% CI) ^3^	*p*-Value ^3^	
SEAMS [pointwise] ^1^	0.93 (0.87–0.99)	**0.026**	0.92 (0.86–0.99)	**0.032**	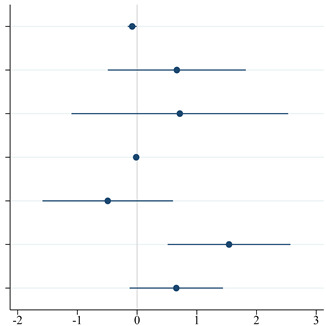
BMQ [ambivalent] ^1,4^	1.90 (0.73–4.97)	0.190	1.95 (0.61–6.18)	0.259	
BMQ [indifferent & skeptical] ^1^	2.31 (0.45–12.00)	0.318	2.05 (0.33–12.54)	0.439	
MSIS-29 total [pointwise] ^1^	1.01 (0.98–1.03)	0.668	0.98 (0.95–1.02)	0.344	
Sex [female]	0.77 (0.32–1.82)	0.547	0.61 (0.21–1.82)	0.378	
Previous DMT [yes]	3.21 (1.30–7.91)	**0.011**	4.66 (1.67–13.04)	**0.003**	
Difference of EDSS [pointwise] ^1^	1.27 (0.67–2.41)	0.465	1.93 (0.88–4.21)	0.100	

^1^ Variables at visit 3 (after 9 months of treatment). The difference of EDSS means the difference between baseline and visit 3. ^2^ Unadjusted represents univariate models separately for each variable; adjusted represents the final multivariate model. For categorical variables, the reference values are these with OR = 1. ^3^ Odds Ratio (OR), Confidence Interval (CI), statistically significant *p*-values are in bold (<0.05). ^4^ vs. BMQ category [accepting].

**Table 3 pharmaceuticals-15-01248-t003:** Clinical and patient-reported outcomes after 3 and 9 months of teriflunomide therapy (N = 105) ^1^.

Clinical Outcomes ^2^	Baseline	Visit 2	Visit 3
Mean Length of Follow-Up	Teriflunomide Initiation	2.9 ± 0.25 months	8.8 ± 0.5 months
Number of patients with at least 1 relapse (%)	66 (62.9%)	4 (3.8%)	8 (7.6%)
Number of patients hospitalized for relapse (%)	31 (29.5%)	0 (0.0%)	1 (1.0%)
ARR	0.72 (0.57–0.91)	0.19 (0.12–0.29)	0.15 (0.09–0.25)
ARR relative decrease	-	73.7%	78.9%
EDSS	1.97 (±0.99)	1.97 (±1.04)	2.03 (±0.99)
MSIS-29 ^3^	26.62 (±20.40)	25.58 (±20.10)	25.49 (±19.88)
MMAS-8 ^3^	-	7.61 (±0.82)	7.39 (±1.22)
VAS ^3^	-	96.21 (±14.52)	95.60 (±14.22)
SEAMS ^3^	-	-	28.53 (±7.20)
BMQ accepting ^3^	-	-	60 (60.6%)
BMQ ambivalent ^3^	-	-	31 (31.3%)
BMQ indifferent ^3^	-	-	6 (6.1%)
BMQ skeptical ^3^	-	-	2 (2.0%)
Patient preferences (oral dosage) ^3^	101 (98.1%)	104 (100.0%)	104 (100.0%)

^1^ Baseline outcomes assessed 12 months before the teriflunomide treatment; at visit 2, a period of 3 months was evaluated (between the baseline and visit 2) and at visit 3, a period of 6 months was evaluated (between visit 2 and 3). ^2^ Values are number (%)/%/mean (± SD)/mean (95% CI). ^3^ Number of patients at visits. MSIS-29: N = 104, N = 105, and N = 104; MMAS-8: N = 103 and N = 104; VAS: N = 90 and N = 97; SEAMS: N = 89; BMQ: N = 99; preferences: N = 103, N = 104, and N = 104.

## Data Availability

Data is contained within the article and Appendix A.

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
