# Peer review of "Relationship between Patient Preferences, Attitudes to Treatment, Adherence, and Quality of Life in New Users of Teriflunomide"

_pharmaceuticals, 2022, doi:10.3390/ph15101248_

Round 1

Reviewer 1 Report

This reviewer would like to have a more clear explanation of the statement "...for the appropriate medication" (line 44 in the Abstract and lines 270 and 271).

Teriflunomide has shown in the published pivotal phase 3 trials low effect in the Annualized Relapse Rate comparison to "high efficacy" medications hence, Aubagio is generally considered as a DMT in the "low-to-medium" efficacy level. If you agree, I wonder if a comment in the discussion section would be in order.    

Author Response

COMMENT 1: This reviewer would like to have a more clear explanation of the statement "...for the appropriate medication" (line 44 in the Abstract and lines 270 and 271).

ANSWER: Thank you for the review and stimulating comments. The wording “appropriate” comes from the original article by Risser et al. (2007; https://pubmed.ncbi.nlm.nih.gov/18232619/) who developed and validated the questionnaire called "Self-Efficacy for Appropriate Medication Use Scale" (SEAMS). We understood from your comment that the wording “appropriate medication” may be confusing. We thus included the definition of “self-efficacy of appropriate medication” from the Risser´s article in all instances throughout the text as follows:

“the confidence in ability to take medications” line 27 (abstract)

“(i.e., the confidence in one’s ability to take medication correctly).” Line 45 (abstract)

“The self-reported confidence in one’s ability to perform a given task such as taking one’s medications” line 129 (body text)

“(i.e., the confidence in one’s ability to take medication correctly)” line 174 (body text)

“(i.e., the self-reported confidence in one’s ability to take medication correctly).” Line 303 (body text)

COMMENT 2: Teriflunomide has shown in the published pivotal phase 3 trials low effect in the Annualized Relapse Rate comparison to "high efficacy" medications hence, Aubagio is generally considered as a DMT in the "low-to-medium" efficacy level. If you agree, I wonder if a comment in the discussion section would be in order. 

ANSWER: Based on your remark, we found and studied the recent network meta-analysis by Li et al. (2021, https://pubmed.ncbi.nlm.nih.gov/33878488/) and indeed, the ARR reduction with teriflunomide can be considered a medium effect compared to other, notably more recent, DMTs.

We add a sentence and a new reference “Although teriflunomide is generally considered a DMT in the "medium" efficacy level,”, line 249 (body text).

Reviewer 2 Report

1. most of the patients treated with teriflunomide as second-line DMD have been previously treated with IFN or glatiramer acetate that are injectable DMD. in paragraph 19 authors declared that." the increasing trend in disease-specific quality of life was most pronounced in the previous DMT group". Then in paragraph 20 they wrote: "patients who were non-adherent after nine months of treatment with teriflunomide had lower disease specific quality of life compared to adherent patients. Therefore I expect that the previous DMT group, who have increasing quality of life, should also be more adherent. This is in contrast with results described in paragraph 16 where decreasing adherence was observed in patients wityh other DMT preceding teriflunomide. Please reconcile or discuss the confliting results. 

In addiction, patients with "needle fatigue" or who expericed toxicity from previous DMT, should have increased quality of life swthching to a oral medication wth an overall good safety. Why they are non adherent? Could this be due to daily administration of oral drug that could be difficult to remember in a patient used to weekly administraion of injectable drug?

Author Response

COMMENT 1: most of the patients treated with teriflunomide as second-line DMD have been previously treated with IFN or glatiramer acetate that are injectable DMD. in paragraph 19 authors declared that." the increasing trend in disease-specific quality of life was most pronounced in the previous DMT group". Then in paragraph 20 they wrote: "patients who were non-adherent after nine months of treatment with teriflunomide had lower disease specific quality of life compared to adherent patients. Therefore I expect that the previous DMT group, who have increasing quality of life, should also be more adherent. This is in contrast with results described in paragraph 16 where decreasing adherence was observed in patients wityh other DMT preceding teriflunomide. Please reconcile or discuss the confliting results.

ANSWER: Thank you for raising this point. Firstly, the previously treated had significantly worse quality of life (higher MSIS-29) in all three visits with p=0.035, p=0.006 and p=0.038 in the first, second, and third visits, respectively. To show this, we add Supplementary Table 6. It is true that the psychological sub-scale in previously treated was the only variable showing a consistent trend of decreasing MSIS-29 through the three visits (we add supplementary Figure 1b). On the other hand, even with this improvement, the pre-treated in visit 3 by far did not reach the quality of life in treatment naïve patients.

In other words, the previously treated had worse point quality of life in all three visits but there was a trend of improvement in their psychological subscale in time. Conversely, the naive patients experienced, on average, no improvement in time but their quality of life was better in all three time points.

To summarize, we propose to add supplementary Table 6 and Figure 1b and to reword the part of the manuscript as follows:

“The increasing trend in disease-specific quality of life was most pronounced in the previous DMT group, notably the psychological subscale. These patients, however, had already a significantly worse quality of life at baseline (p=0.035) and thus did not reach the MSIS-29 score of the naïve patients even after improvement in the second (p=0.006) or the third visit (p=0.038) (Supplementary Table 6).”, lines 210 to 214

“Patients who were non-adherent after nine months of treatment with teriflunomide (MMAS-8 ≤ 6 points) had, on average, a lower disease-specific quality of life score compared to adherent patients (MMAS-8 > 6 points) irrespective of the previous therapy.”, lines 225 to 227

We also propose to clarify the sentence in line 172-173 as follows: “Also, there was a drop in adherence observed with each other line of therapy preceding teriflunomide (details not presented).”

COMMENT 2: In addiction, patients with "needle fatigue" or who expericed toxicity from previous DMT, should have increased quality of life swthching to a oral medication wth an overall good safety. Why they are non adherent? Could this be due to daily administration of oral drug that could be difficult to remember in a patient used to weekly administraion of injectable drug?

ANSWER: It is true that it remains unexplained by our study why the switchers had worse adherence. A possible explanation, as you suggested, could be that they were not used to daily administration of the DMT. To verify that, we tested the difference in SEAMS scores between the switchers and treatment naïve patients at baseline and the third visit but found no significant difference.

We add the following text into the discussion section “Contrarily to our results, one would expect the patients switching from previous therapies to be more adherent to teriflunomide which is a safe and efficacious DMT. A possible explanation of the non-adherence in the second line could be the need for the daily administration of oral drugs being difficult to remember, especially in a patient used to weekly administration of an injectable drug. On the other hand, the SEAMS score was not significantly different between switchers and naïve patients at baseline (p=0.09) nor the third visit (p=0.465). After all, the reasons for worse adherence in switcher remains unexplained by our study.”, lines 286 to 293.

Round 2

Reviewer 2 Report

Thank you for addressing the point raised during my revision. 

As an outlook, the study may be improved in the future by including a second arm in which the same paramenters are collected in patients taking a different oral drug indicated in second line treatment of MS (i.e dimethyl fumarate). This may explain if lower adherence is due to oral administration or other factors linked to teriflunomide. 

Author Response

Thank you for this idea. It is important to suggest a way forward for future research. We believe the sentence will aptly add to the discussion, more specifically, lines 280 to 284 of the enclosed manuscript. 
